# Robust Quad-Tree based Registration on Whole Slide Images

**Christian Marzahl**                                              christian.marzahl@fau.de
*Pattern Recognition Lab, Computer Sciences Friedrich-Alexander-Universität Erlangen-Nürnberg Erlangen, Germany*

**Frauke Wilm**
*Pattern Recognition Lab, Computer Sciences Friedrich-Alexander-Universität Erlangen-Nürnberg Erlangen, Germany*

**Christine Kröger**
*EUROIMMUN Medizinische Labordiagnostika AG Lübeck, Germany*

**Franz F. Dressler**
*Institute of Pathology University Medical Center Schleswig-Holstein Campus Lübeck, Germany*

**Lars Tharun**
*Institute of Pathology University Medical Center Schleswig-Holstein Campus Lübeck, Germany*

**Sven Perner**
*Institute of Pathology University Medical Center Schleswig-Holstein Campus Lübeck, Germany*

**Christof A. Bertram**
*Institute of Pathology University of Veterinary Medicine Vienna, Austria*

**Jörn Voigt**
*EUROIMMUN Medizinische Labordiagnostika AG Lübeck, Germany*

**Robert Klopfleisch**
*Institute of Veterinary Pathology Freie Universität Berlin Berlin, Germany*

**Andreas Maier**
*Pattern Recognition Lab, Computer Sciences Friedrich-Alexander-Universität Erlangen-Nürnberg Erlangen, Germany*

**Marc Aubreville**
*Technische Hochschule Ingolstadt Ingolstadt, Germany*

**Katharina Breininger**
*Department of Artifical Intelligence in Biomedical Engineering Friedrich-Alexander-Universität Erlangen-Nürnberg Erlangen, Germany*

**Editor:**

## Abstract

The registration of whole slide images (WSIs) provides the basis for many subsequent processing steps in digital pathology. For instance, the registration of immunohistochemistry (IHC) and hematoxylin & eosin (H&E)-stained WSIs is usually the first step in guiding IHC diagnostic procedures. Still, many registration methods operate poorly on WSIs. Reasons for this include the WSI size, fluctuating image quality or elastic tissue deformations. Multiple prior methods are further specialised towards a specific image modality, such as histology or cytology, or rely on a specific preparation protocol. To minimise these ef-

fects, we developed a robust WSI registration, which differs from previous methods by the following new aspect: We introduce a multi-scale approach based on a quad-tree (QT), with several termination criteria that makes the algorithm particularly insensitive to tissue artefacts and that further allows to estimate a piece-wise affine transformation. We validated our method on five scanner systems and 60 WSIs with different stainings. Our results outperformed any publicly available WSI registration method. The QT code, WSI landmarks and tools used to create the validation dataset are made publicly available.

**Keywords:** Registration, Microscopy, Pathology

## 1. Introduction

Digital pathology has experienced an unprecedented growth in recent years due to the availability of scanners with Food and Drug Administration (FDA) clearance, progress in software applications as well as an increasing acceptance by pathologists (Aeffner et al., 2019). The open challenges in digital pathology include scanner independence for automatic analysis algorithms as well as the simultaneous assessment of multiple stainings. For the first case, datasets where the same slide is digitised by multiple scanners play an essential role. To avoid an extremely time-consuming re-annotation and instead transfer the labels from one scanner to the next, fine-grained alignment of the WSIs is essential. In the second case, standard H&E datasets are often complemented by specialised IHC reactions that allow for a more detailed assessment of different pathologies (see Sec. 2.1). Aligning the different visualisations allows both to transfer annotations from one staining to another and to jointly visualise the same image region in both WSIs, supporting a faster assessment by the pathologist.

For these purposes, robust WSIs registration techniques are required. Proprietary WSI formats by different scanner vendors, differing magnification factors, multi-resolution pyramid image storage, background representation and focus algorithms make WSI registration a challenging task. Apart from these technical aspects, the samples themselves can contain artefacts like tissue detachment, deformation, folding or tearing, which poses additional challenges for any registration method.

Registration plays a significant role for many medical image processing tasks across different imaging modalities (Song et al., 2017). Two main classes of registration approaches are intensity- and keypoint-based methods. Intensity-based methods attempt to minimise the intensity gradient between a moving (source) and a fixed (target) image (Jiang et al., 2019); In contrast, keypoint-based strategies use keypoint extractors like Oriented FAST and Rotated BRIEF (ORB) (Rublee et al., 2011) or Scale Invariant Feature Transform (SIFT) (Lowe, 2004) and then match and align corresponding keypoints in multiple images. In both cases, the extracted information is used to calculate a linear transformation (translation, rotation, skewing, scaling) or a non-linear transformation that compensates for non-linear distortions between the source and target WSI. An example of a rigid transformation is the registration of the same WSI scanned with multiple scanners, while a non-rigid registration is needed for registration of consecutive slides with or without different stains. In addition to the methods of classical image processing, deep learning-based methods can also be applied for registration tasks (Lee et al., 2019) as a whole or parts thereof.

In the recent Automatic Non-Rigid Histological Image Registration (ANHIR) benchmark challenge (Borovec et al., 2020), several algorithms have been proposed for non-linear

registration; however, the submitted approaches do not naively support WSIs and depend upon down-sampled images (down to 2%) and therefore would require considerable modifications to operate on full WSIs. A further limitation of the ANHIR dataset is the use of the same source and target scanner, which neglects particular challenges like different scanned areas, magnifications and resolution levels. Furthermore, pathological registration methods often require a form of tissue pre-segmentation (Levy et al., 2020; Jiang et al., 2019; Rossetti et al., 2017) or background knowledge about anatomical structures present in the images (Schwier et al., 2013). For cytology WSIs, which where not included in the ANHIR dataset, these approaches are not applicable, since they - unlike tissue-based WSIs - do not contain contiguous areas for segmentation. Additionally, the down-sampling of WSIs could lead to the disappearance of small space structures like cells and therefore prevent a successful registration. Out of the mentioned methods, only Jiang et al. (2019) provide publicly accessible source code and are able to deal with WSI images directly, but the approach is limited to a translation-based registration that ignores rotation or scaling differences.

In this paper, we therefore propose a general and configurable approach based on a quad-tree (QT) (Bardera et al., 2006) that supports a robust and fast piece-wise affine (rigid + scaling) registration between full-scale WSI for many application fields out of the box. No additional preconditions like a pre-segmentation or a specific anatomical structure have to be fulfilled. We validate our method with manually positioned landmarks on 60 WSIs from four different fields of application across different scanners and compare the results with the open-source WSI-registration method by Jiang et al. and the publicly available methods from the ANHIR challenge on down-sampled sub-images. To ensure the reproducibility of our results, we publish the code for our method as Pip Python package[1] together with the evaluation code[2] and implementation[3]. The WSIs can be provided by the corresponding author upon reasonable request.

## 2. Materials

For our experiments, we compiled four WSI datasets which cover a broad range of WSI registration tasks. Two datasets address registration of the same slide digitised with four different scanners and contain five H&E-stained tissue slides and five Prussian Blue pulmonary hemorrhage cytology WSIs, respectively. The other two datasets allow to assess the performance between different stains, namely H&E and IHC. First, the *same* five tissue slides were stained and digitised, then destained, IHC-stained and again digitised with the same scanner. Secondly, five H&E and five *consecutive* IHC slides were digitised with two different scanners. The datasets and the ground truth generation are described in more detail in the following sections.

### 2.1 Datasets

**Multi scanner cytology dataset (MSC):** For the cytology WSI dataset, we used five equine bronchoalveolar lavage fluid samples. The samples were prepared by cytocentrifuga-

---

1. https://pypi.org/project/qt-wsi-registration/
2. https://github.com/ChristianMarzahl/EIPH_WSI
3. https://github.com/ChristianMarzahl/WsiRegistration

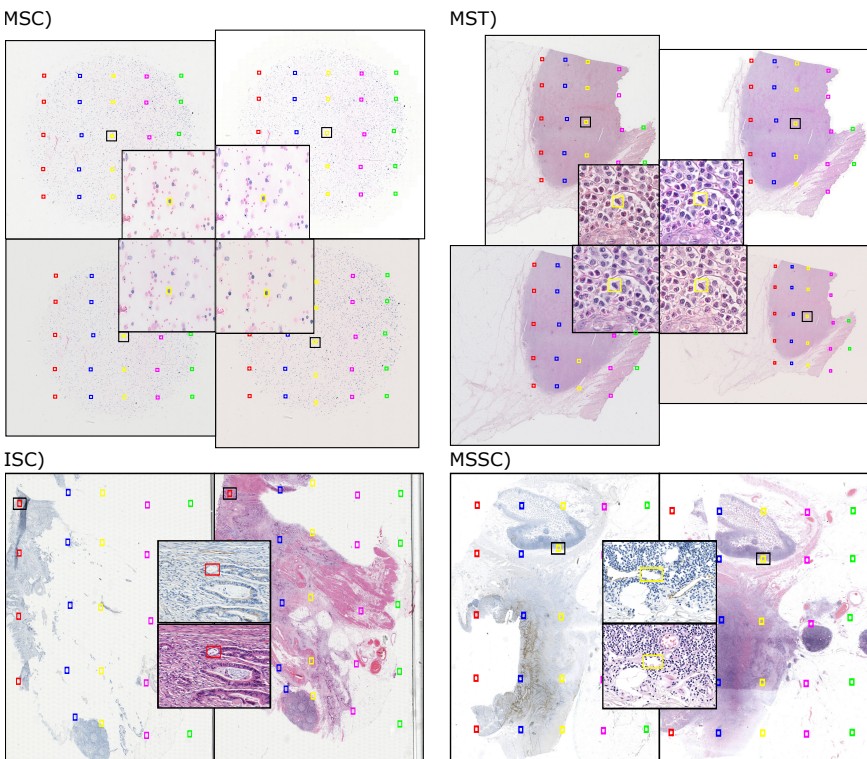

Figure 1: The top row represents one WSI of the MSC and one of the MST dataset, digitised with four different scanners. The bottom row depicts the ISC dataset (C) where one slide is stained with H&E, destained and re-stained with IHC in combination with severe tissue ablation artefacts. Subfigure (D) shows two consecutive slices from the same specimen of the MSSC dataset where one is IHC- and one is H&E-stained. The red, blue, yellow, purple and green rectangles depict the manually defined landmarks.

tion and stained with Prussian Blue. Digitisation of the glass slide was performed using four different linear scanners. The Aperio ScanScope CS2 (WSI resolution 0.2533 $\frac{\mu m}{px}$), Hamamatsu NanoZoomer S210 (WSI resolution 0.2533 $\frac{\mu m}{px}$) and the Hamamatsu NanoZoomer 2.0-HT (WSI resolution 0.2533 $\frac{\mu m}{px}$) each with a magnification of 400× and the ZEISS Axio Scan.Z1 at a magnification of 200× (WSI resolution 0.221 $\frac{\mu m}{px}$). The digitisation process resulted in 20 WSI and in the following we will refer to this dataset as MSC.

**Multi scanner tissue dataset (MST):** For the H&E tissue dataset, we used slides from canine cutaneous mast cell tumours, also digitised with the same four scanners as the MSC dataset. We will refer to this dataset as MST.

**In-situ proteomic single scanner dataset (ISC):** In pathological diagnostics, IHC-staining is used to visualise expression patterns of so-called marker proteins that are expressed only by certain cell or tissue types. This allows to detect structures that cannot

be discriminated by H&E morphology alone. D2-40 is such a marker protein, specifically expressed by cell linings of lymphatic vessels. These thin-walled vessels are especially relevant in cancer metastasis as they become invaded by tumor cells, which thus spread to lymph nodes and other organs. To include this diagnostically highly relevant information, we first H&E-stained five colorectal cancer tissue samples from five different patients to obtain morphological information. After digitisation with a Roche Ventana iScan HT scanner with $200\times$ magnification (WSI resolution 0.25 $\frac{\mu m}{px}$), the cover slips were removed and the sections unstained, washed, subjected to IHC for the lymphatic vessel marker D2-40 and digitised again. The resulting dataset of paired morphological and *in-situ* D2-40 expression data is referred to as ISC dataset.

**Multi-slide multi scanner dataset (MSSC):** The procedure of staining, destaining and IHC-restaining of the same slide has potential disadvantages like physical trauma to the slide tissue and a reduced final staining quality. An alternative is the use of two consecutive slides where one slide is IHC-stained and the second slide is H&E-stained. We prepared this dataset from the same samples and in analogy to ISC, and reference it as MSSC.

## 2.2 Ground truth generation

To validate the registration accuracy, we placed a 5x5 rectangular grid of landmarks on each WSI such that it covers as much tissue as possible (see Fig. 1) using the online annotation software EXACT (Marzahl et al., 2021). We manually refined the position of each landmark on the highest resolution level such that they are positioned on the same cell or prominent tissue landmarks near-by in corresponding WSIs. This landmark pattern allows us to measure the pixel-level deviation at 25 locations on the WSI and to quantify any macroscopic elastic deformation. In total, we created and manually checked 1500 registration points on 60 WSIs. The landmarks are sized according to representative registration target structures for the individual datasets, e.g. mitotic figures for MST, hemosiderophages for MSC and lymphatic vessels for ISC and MSSC. Thereby, it can be determined whether the registration results are accurate enough to be used as ground truth for object detection algorithms.

## 3. Methods

The goal of any image registration method is to find a transformation $T$ between a moving source image $I$ and a fixed target image $I'$. For the datasets MSC and MST the same slides were digitised with multiple scanners; thus, we assume that an affine transformation is suitable to estimate the registration. For the ISC and MSSC dataset, a non-rigid registration is required to handle tissue distortions introduced by the use of sequential tissue slides or the process of de- and re-staining. To handle both use cases, we construct our registration pipeline as follows: Based on a QT backbone, we recursively divide the WSI into image segments with successively higher resolution levels. For each segment, we estimate a transformation matrix based on matching keypoints in both images. We thereby refine the estimate from the previous QT level until a previously defined termination criterion is reached. This results in a piece-wise affine approximation of any non-linear deformation.

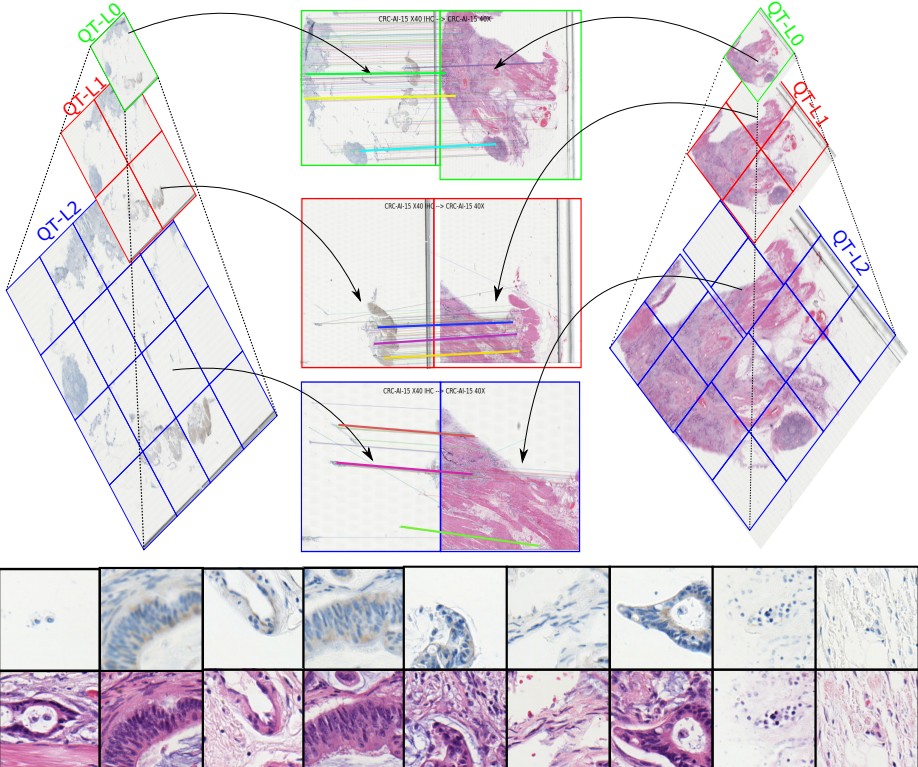

Figure 2: An example WSI with an IHC (left) and H&E (right) staining from the ISC dataset with severe tissue artefacts. The figure depicts three QT levels (green, red, blue rectangles) on the source slide (left) and their transformed counterparts on the target slide (right). The center shows an example patch for each level with matched keypoints depicted as lines. The bottom line shows patches centered on matched keypoints.

As keypoint detector, we use the low-level extractor SIFT. In the following section, we will describe our method in detail.

### 3.1 Quad tree-based registration approach

QTs are characterised by their recursive definition, where each segment on the previous level is divided into four sub-segments that have twice the spacial resolution but cover only a quarter of the spacial extent (see Fig. 2). Their hierarchical structure makes them ideal candidates to approximate WSIs with their pyramidal approach of saving images at multiple magnification levels. Using this data structure as the backbone, our registration algorithm is recursively applied for each level of the QT and can be described as follows: First, we extract a source and a target region of interest (RoI) from the source/target WSI. On QT level 0, this RoI is simply the complete WSI, whereas it consists of a sub-segment in subsequent levels of the QT. In the second step, keypoints and their respective feature vectors are

extracted from source and target RoIs using SIFT. Then, corresponding keypoints in both RoIs are determined using k-nearest neighbours (k=2) on the feature vectors. To avoid that outliers impact the estimation negatively, the point pairs are filtered as described by Lowe (2004). For a total number of $n$ successfully matched keypoints, $\vec{x}_i \in \mathbb{R}^2, i \in [1, n]$ will refer to the pixel coordinates of a keypoint $i$ in the source image, and $\vec{x}'_i \in \mathbb{R}^2$ to the position of the matched keypoint in the target image. In the next step, we compute the transformation matrix $T \in \mathbb{R}^{2 \times 3}$ with eight degrees of freedom by computing the homography matrix (OPENCV implementation) with RANSAC (Fischler and Bolles, 1981) filtering or Coherent Point Drift (Myronenko and Song, 2010) depending which resulted in a lower error between matched keypoints. This keypoint matching error (KME) $E$ is the mean Euclidean distance between the transformed source keypoints $T(\vec{x}_i)$ and the target keypoints $\vec{x}'_i$ and is defined as

$$E = \frac{1}{n} \sum_{i}^{n} ||\vec{x}'_i - T(\vec{x}_i)||_2 \tag{1}$$

The final step is the evaluation of the QT termination criteria. The implementation allows to define flexible stopping criteria based on maximum QT depth, run-time, a user-defined threshold for the KME or when the KME increases compared to the previous level. To keep the evaluation consistent, a maximum QT level of 2 was used as the stopping criterion for the results presented here. If the termination criterion is not fulfilled, the source RoI is divided into four equal parts. The calculated $T$ is applied to each of the 16 corner points to estimate the corresponding RoIs on the target slide and the next QT level with the source and target RoI is initialised recursively. To calculate the position of a given point in the source image after registration, the final transformation matrix associated with the last QT level is used to transform the point to the target coordinate system.

## 4. Results

We evaluate our registration results by calculating the mean target registration error (TRE) in pixels (analogous to Eq. 1) between the ground truth landmarks and the transformed landmarks. For comparability between scanners with different microns per pixel (MPP) values, all values are converted to microns. Furthermore, we calculate the mean Intersection over Union (IoU) over all landmarks to quantify if the registration algorithm is accurate enough for cell-level registration. For comparison, we use the open-source implementation of Jiang et al. (2019), which we will refer to as ReStain. Because the provided implementation has no compensation for different MPP values, we re-scaled all target WSIs according to the MPP values read out from the meta data provided by the scanner. Furthermore, we use the publicly available methods from the the ANHIR challenge (Borovec et al., 2020): DROP, ANTs, Elastix, NiftyReg, bUnwarpJ which are provided as docker containers [4]. The methods are described in detail in Borovec et al. (2020) and due to their inability to handle full WSI we down-sampled all image to $4k \times 4k$ as described in Borovec et al. (2020). As additional reference, we calculate an affine matrix between the source and corresponding target landmarks to estimate the error of a least-squares optimal affine registration. We further report the initial mean Euclidean pixel distance without registration.

---

4. https://github.com/Borda/BIRL

| method | MST | | MSC | | ISC | | MSSC | | Time | WSI |
|---|---|---|---|---|---|---|---|---|---|---|
| | TRE | IoU | TRE | IoU | TRE | IoU | TRE | IoU | sec. | support |
| Initial | 3529 | .00 | 1507 | .00 | 1508 | .00 | 4250 | .00 | - | - |
| LM | **2** | **.80** | 1 | **.91** | 6 | .80 | 1508 | .04 | - | - |
| QT_L0 | 7 | .45 | 4 | .77 | 16 | .57 | 613 | .01 | 2 | Y |
| QT_L1 | 3 | .65 | 2 | .87 | 4 | .84 | 449 | .01 | 19 | Y |
| QT_L2 | **2** | .79 | **2** | .88 | **3** | **.87** | **416** | **.04** | 134 | Y |
| ReStain | 237 | .00 | - | - | 44 | .00 | 1178 | .00 | - | Y |
| ANTs | 905 | .00 | 586 | .00 | 1127 | .00 | 2691 | .00 | **1** | N |
| DROP | 961 | .00 | 591 | .00 | 1255 | .00 | 2657 | .00 | 18 | N |
| Elastix | 555 | .00 | 251 | .00 | 212 | .00 | 927 | .00 | 50 | N |
| NiftyReg | 634 | .17 | 460 | .04 | 1212 | .01 | 1961 | .00 | 26 | N |
| bUnwarpJ | 28 | .02 | 373 | .06 | 636 | .03 | 1051 | .00 | 33 | N |

Table 1: Registration results for all datasets and methods. ReStain (Jiang et al., 2019), and DROP, ANTs, Elastix, NiftyReg, bUnwarpJ refer to the ANHIR challenge methods, QT(L0,L1,L2) represent the QT levels and LM the affine transformation between the source and target landmarks and initial the transfer of the ground truth landmarks. TRE represents the target registration error between the target landmarks and transformed source coordinates in microns, followed by the IoU, mean runtime in seconds (sec) and if the method has a native WSI support.

### 4.1 CMS and MST dataset

For the MSC and MST datasets, we use the Aperio scanner as source and the remaining three scanners as targets. The QT configuration with the best results uses SIFT with 1024 keypoints and an RoI size of $1024 \times 1024$ pixels. As shown in Table 1, each level of the QT reduces the pixel distance error between the ground truth landmarks and the transformed landmarks until, at level two, the QT results outperform the affine landmark transformation. We were unable to evaluate the ReStain approach on the MSC dataset due to the method's dependence on an initial segmentation which we could not parameterise with the provided options.

For the ISC and MSSC dataset, we use the IHC WSIs as registration source. For the ISC dataset, we see equally high performance as for MSC and MST. In contrast, for the MSSC dataset the QT-based method is not able to match the error of the affine landmark transformation. We attribute this to the lack of matching (SIFT) keypoints since slides of sequential slices of tissue are registered: Compared to a mean of 268 matching keypoints on the ISC dataset, we only found a mean of 13 keypoints on the MSSC dataset. All quantitative results including the runtime can be found in Tab. 1.

## 5. Discussion and Conclusion

The registration of WSIs is a very challenging task due to their size and further difficulties like different stainings, scanners and image artefacts. We show that our method produces

promising results on three out of four datasets and outperforms the reference method in all cases. Furthermore, we offer the first registration method that is able to register cytology WSIs as it does not require any kind of pre-segmentation. The importance of native WSI-based approaches is highlighted by both methods outperforming down-sampling strategies. With the proposed approach, the results achieved for the different QT levels are considerably below the resolution level used in the respective RoI. We hypothesise that this is due to the large number of keypoints extracted in each RoI that allow for a sub-pixel accurate estimation of the transformation matrix. The fact that we are able to outperform an affine transformation based on point correspondences can be attributed to, e.g. stitching artefacts. One limitation of our work is highlighted when applied to consecutive tissue slices on which SIFT only finds a limited number of matching keypoints. It is striking that all approaches performed abysmally on this data. We reason that the use of different source and target scanners and heavy tissue artefacts particularly challenge registration. Still, this opens up the possibility of future improvements by implementing a specifically trained deep-learning-based keypoint detector as an alternative or to integrate intensity-based methods as well. In this paper, we showed that the proposed QT-based approach is accurate, efficient and robust for WSI registration. These properties are underlined by the fact that the proposed registration method is utilized in the context of a detection challenge. Due to its flexibility, we see a high potential to extend this method to further applications, e.g., as robust initialisation for fully elastic registration approaches.

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
