# OpenReview forum: "Robust quad-tree based registration of whole slide images"
_MICCAI.org/2021/Workshop/COMPAY — COMPAY 2021_

### Official Review · Reviewer_tQ8u · 2021-08-03
**This is an interesting methods paper that deal with the issue of image registration in pathology slides**

**Rating:** 7
**Confidence:** 2

**Review:**

This is an interesting methods paper that deal with the issue of image registration in pathology slides. The authors propose a novel approach based on quad-trees, which in this setting provide a piece-wise transformation at increasing levels of magnification. In various real-life scenarios (same slide with H&E and IHC) they show that their method works well. On another realistic example (sequential slide H&E and IHC), their method does not perform well. The authors link this issue to a true structural difference between the two slides that causes the SIFT keypoints to differ.

The paper is well written and structured clearly.

The paper contains anonymized links to github and pypi to share the code. The links should be checked and made available before publication.

---

### Official Review · Reviewer_qBsV · 2021-08-16
**Interesting problem but some important details are missing**

**Rating:** 7
**Confidence:** 4

**Review:**

This paper presents a whole slide image (WSI) registration method based on a multi-scale approach based on a quad-tree (QT). The proposed algorithm used multiple termination criteria to overcome insensitivity introduced by tissue artifacts. The study was performed on multiple data sets containing 60 WSIs using different stainings and five different scanners. The data including QT code and WSI landmarks used to create the validation dataset will be publicly released.

Overall, the manuscript is well structured and approaches an interesting problem. However, below are some major/minor comments
- Adding a flow diagram describing the main components of the proposed algorithm would help better understanding this approach
- Ablation study is missing, it is not clear how the approach work without QT or with  octree and trying with different stopping criteria
- Qualitative results are missing which makes it difficult to understand the impact of the algorithm visually
- It is not clear how the k-nearest neighbors (k=2) and other hyperparameters are selected and the intuition behind the 5x5 rectangular grid and 1500 registration points from 60 images?
- In the abstract, it is stated that ‘Our results outperformed any publicly available WSI registration method’ but this claim is not well justified in the manuscript

---

### Decision · Program_Chairs · 2021-08-25

Accept